# Characterization of Fructose-1,6-Bisphosphate Aldolase 1 of *Echinococcus multilocularis*

**DOI:** 10.3390/vetsci9010004

**Published:** 2021-12-23

**Authors:** Xuedong He, Jing Zhang, Yue Sun, Tianyan Lan, Xiaola Guo, Xiaoqiang Wang, Omnia M. Kandil, Mazhar Ayaz, Xuenong Luo, Houhui Song, Yadong Zheng

**Affiliations:** 1Key Laboratory of Applied Technology on Green-Eco Healthy Animal Husbandry of Zhejiang Province, Zhejiang Provincial Engineering Laboratory for Animal Health Inspection and Internet Technology, College of Animal Science and Technology and College of Veterinary Medicine, Zhejiang A&F University, Hangzhou 311300, China; hexuedong2547@126.com; 2National Institute of Parasitic Diseases, Chinese Center for Disease Control and Prevention, Key Laboratory of Parasite and Vector Biology, Ministry of Health, National Center for International Research on Tropical Diseases, WHO Collaborating Center for Tropical Diseases, Shanghai 200025, China; silkfan@126.com; 3State Key Laboratory of Veterinary Etiological Biology, Key Laboratory of Veterinary Parasitology of Gansu Province, Lanzhou Veterinary Research Institute, CAAS, Lanzhou 730046, China; sunyue1103@126.com (Y.S.); lantianyanvip@126.com (T.L.); guoxiaola@caas.cn (X.G.); luoxuenong@caas.cn (X.L.); 4School of Chemical and Biological Engineering, Lanzhou Jiaotong University, Lanzhou 730070, China; wangxiaoqiang@lzjtu.edu.cn; 5National Research Centre, Departement of Parasitology and Animal Disease, Veterinary Research Division, Giza 12622, Egypt; kandil_om@yahoo.com; 6Department of Parasitology, Cholistan University of Veterinary and Animal Sciences, Bahawalpur 73000, Pakistan; mazharayaz@cuvas.edu.pk

**Keywords:** *Echinococcus multilocularis*, fructose-1,6-bisphosphate aldolase 1, secondary structure, circular dichroism, enzyme kinetics

## Abstract

Glycolysis is one of the important ways by which *Echinococcus multilocularis* acquires energy. Fructose-1, 6-bisphosphate aldolase (FBA) plays an important role in this process, but it is not fully characterized in *E. multilocularis* yet. The results of genome-wide analysis showed that the *Echinococcus* species contained four *fba* genes (FBA1-4), all of which had the domain of FBA I and multiple conserved active sites. EmFBA1 was mainly located in the germinal layer and the posterior of the protoscolex. The enzyme activity of EmFBA1 was 67.42 U/mg with *K*_m_ and *V*_max_ of 1.75 mM and 0.5 mmol/min, respectively. EmFBA1 was only susceptible to Fe^3+^ but not to the other four ions (Na^+^, Ca^2+^, K^+^, Mg^2+^), and its enzyme activity was remarkably lost in the presence of 0.5 mM Fe^3+^. The current study reveals the biochemical characters of EmFBA1 and is informative for further investigation of its role in the glycolysis in *E. multilocularis*.

## 1. Introduction

*Echinococcus multilocularis*, the causative agent of alveolar echinococcosis (AE), is widely distributed in the Northern hemisphere [1], and is mainly found in Tibet, Qinghai, Gansu and other high-altitude areas in China [2]. In the life cycle, the adult worm resides in the intestine of canids (mainly foxes) acting as a definitive host and the eggs expelled with the feces contaminate food and water, which may cause rodents and occasionally humans to be infected. Then the eggs develop into the metacestodes in the liver and lung. If foxes prey on these infected rodents, the metacestodes grow into the adults, thus finishing an entire life cycle. In an intermediate host, *E. multilocularis* displays a tumor-like infiltrative growth. The latent period of AE is long, up to five years, and the approaches for AE treatment are very limited. Clinically, parasites can be removed by surgery with a high risk of secondary infection and serious immune responses due to cyst fracture [3,4]. It was estimated that the mean annual incidence of AE was approximately 0.26 per 100,000 population in Switzerland, with mortality of >90% within 10 to 15 years after diagnosis in patients with or without proper treatment [5].

The energy metabolism of parasites mainly depends on the Embden-Meyerhof-Parnas pathway (EMP) [6,7]. Fructose-1, 6-bisphosphate aldolase (FBA) is a key enzyme that is involved in the first stage of glucose metabolism, where glucose is broken down into pyruvate. FBA is able to catalyze the breakdown of fructose-1, 6-diphosphate (FDP) into one molecule of dihydroxyacetone phosphate (DHAP) and one molecule of glyceraldehyde-3-phosphate (GAP), and this reaction process is reversible. To date, FBA has been described in many parasites, including *Toxoplasma gondii* [8], *Plasmodium knowlesi* [9], *Schistosoma mansoni* and *Schistosoma japonicum* [10], *Trichinella spiralis* [11] and *E. granulosus* [12]. FBA plays an important role in parasite growth, development, metabolism and substance transport [8,13,14]. FBA not only participates in the glycolytic pathway, but also plays an important role in other processes in parasites. For instance, it was possibly involved in the invasion and motility of *Plasmodium* sporozoites [15,16], and immune evasion during *S. japonicum* infection [17,18]. There is growing evidence to support that FBA can act as a potential drug therapeutic target for parasitic diseases [19]. However, little is known about *E. multilocularis* FBA.

In this study, we conducted the genome-wide analysis of the *fba* genes in the *Echinococcus* species, and analyzed the secondary structure of *E. multilocularis* FBA1 (EmFBA1) and its localization by immunofluorescence assay. Finally, we determined the enzyme kinetics of EmFBA1.

## 2. Materials and Methods

### 2.1. Parasites

The larva of *E. multilocularis* was passaged in *Meriones unguiculatus* in our lab. The cyst mass was aseptically dissected from infected *M. unguiculatus* and washed several times in PBS. Then samples were immediately immersed in 4% paraformaldehyde for preparation of sections or stored at −80 °C for total RNA extraction.

### 2.2. Identification and Phylogenetic Analysis of FBA Genes

The full amino acid sequence of EmFBA1 (EmuJ_000905600) was used to query protein data of *Echinococcus* species (*E. multilocularis* and *Echinococcus granulosus*) in the WormBase ParaSite (https://www.parasite.wormbase.org, accessed on 1 August 2020). For identification of motifs and/or domains, each FBA amino acid sequence was aligned and analyzed using the database of Conserved Domains in NCBI (https://www.ncbi.nlm.nih.gov/Structure/cdd/docs/cdd_search.html, accessed on 1 August 2020).

Alignment of the FBA amino acid sequences was performed using Clustal W (MEGA 7.0, Mega Limited, Auckland, New Zealand) with the default parameter. Before the construction of a phylogenetic tree, an optimal model was selected using Molecular Evolutionary Genetics Analysis (MEGA 7.0, Mega Limited, Auckland, New Zealand). The phylogenetic tree was created by the maximum likelihood method using the WAG model with 1000 bootstrap replications.

### 2.3. RNA Extraction and cDNA Synthesis

Total RNA was extracted using TRIzol reagent (Invitrogen, Carlsbad, CA, USA) according to the method provided by the manufacturer. Briefly, parasites were ground into powder in liquid nitrogen, followed by homogenization in TRIzol, aqueous and organic phase separation using chloroform and RNA precipitation using isopropanol. After centrifugation, RNA was washed in 75% alcohol and dissolved in RNase-free water. The concentration and integrity of the extracted RNA were analyzed by Nanodrop 2000 (ThermoFisher Scientific, Waltham, MA, USA).

1 μg of total RNA was used for cDNA synthesis using RevertAid cDNA First-strand Synthesis Kit (ThermoFisher Scientific) in accordance with the instructions.

### 2.4. Polymerase Chain Reaction and Prokaryotic Expression of EmFBA1

Using the previously synthesized cDNA as a template, a pair of specific primers (Primer Premier 5.0) were used to amplify the open reading frame of *emfba1* gene by PCR: FBA-F: 5′-GGATCCATGTCTCGTTTTGTTCCCTAC-3′ and FBA-R: 5′-GTCGACCCTAGTAGGCATGGTTGGCC-3′. PCR was performed using T100 Thermal Cycler (Bio-Rad, California, USA) with the following steps: 95 °C for 5 min, followed by 35 cycles of 98 °C for 10 s, 55 °C for 30 s, 72 °C for 1 min, and finally 72 °C for 5 min. PCR products were resolved using 1.2% agarose gel and purified by PCR Purification Kit (TSINGKE).

The purified PCR products were subcloned into pET-28(+) (TaKaRa) and transformed into trans-5α competent cells. The positive plasmid was verified by double digestion and sequenced (TSINGKE), and then transformed into BL21 (DE3) competent cells. The positive clone was induced with IPTG (BBI life sciences) at a final concentration of 0.5 mmol/L on 20 °C for 8 h. The collected bacterial cells were treated by ultrasound, centrifuged at 4 °C for 20 min at 10,000× *g*, and the supernatant were used for purification of EmFBA1.

### 2.5. Preparation of Polyclonal Antibodies against EmFBA1

The recombinant EmFBA1 was purified by Ni Sepharose 6 Fast Flow (GE Healthcare, Stockholm, Sweden) and dialyzed by 10K centrifuge filter units (Merck, Branchburg, NJ, USA). The purified protein was resolved using 10% SDS-PAGE gel and analyzed by ImageJ, and its concentration was determined using BCA Protein Assay Kit (Beyotime, Hangzhou, China).

Polyclonal antibodies were produced by immunizing New Zealand white rabbits with emulsified recombinant EmFBA1. First, purified EmFBA1 was mixed with an equal volume of Freund’s complete adjuvant and well emulsified. Then rabbits were subcutaneously injected at multiple locations with emulsified EmFBA1 of 200 μg per rabbit. After 15 days, the second immunization with 200 μg EmFBA1 emulsified with an equal volume of Freund’s incomplete adjuvant was administrated in each rabbit. Finally, the last immunization was performed after 15 days in the same way. Sera were collected and sequentially purified using saturated ammonium sulfate and HiTrap™ protein G (GE Healthcare) according to the manufacturer’s instructions.

### 2.6. Immunofluorescence and Western Blotting

Paraffin sections were dewaxed by xylene, followed by antigen repair in 0.01 M citrate buffer. After wash, the sections were blocked with 5% BSA (Amresco, Pittsburgh, PA, USA). The slides were then incubated overnight at 4 °C with the purified anti-EmFBA1 antibodies diluted at 1:100. After wash, Alexa Fluor 594 goat anti-rabbit antibodies diluted at 1:10,000 (Merck) were added and incubated at room temperature for 1 h, followed by overnight incubation with DAPI (Merck). The slides were observed under fluorescence microscope (Leica, Berlin, Germany).

Western blotting was conducted as previously described [20]. Briefly, 10 μg of the crude tapeworm proteins (*E. multilocularis*, *E. granulosus*, *Taenia hydatigena* and *Taenia asiatica*) stored in our laboratory was resolved using 10% SDS-PAGE gel, and then transferred to PVDF membrane (Millipore, Burlington, MA, USA). The membrane was sequentially incubated with 1:1000 diluted purified anti-EmFBA1 or 1:10,000 diluted anti-acitn (Abcom, London, UK) and then 1:10,000 diluted goat anti-rabbit IgG HRP linked (Sera care, Gaithersburg, MD, USA). After wash, the membrane was dealt with ECL HRP Chemiluminescent Substrate Reagent kit (Invitrogen, Carlsbad, CA, USA) and then exposed to X-ray film (Carestream, Rochester, NY, USA) for visualization. In this experiment, healthy rabbit serum was used as control.

### 2.7. Determination of Enzyme Kinetics

The enzyme activity was determined using FBA Activity Detection Kit (Solarbio, Beijing, China) according to the protocol previously reported [18]. Briefly, EmFBA1 (40 μg/mL) was mixed with the following reagents in the 200 μL reaction system: 2 mM fructose-1, 6-bisphosphate (Sangon Biotech, Shanghai, China), 400 μM NADH (Sigma, River Edge, NJ, USA), 100 μg/mL bovine serum (BSA), 1 unit glycerol phosphate dehydrogenase (Sigma) and 1 unit triose phosphate isomerase (Sigma). Under the conditions of fixed substrate and EmFBA1 concentration, 50 mM Tris-HCl buffer with a different pH from 6.0 to 9.0 and different reaction temperature from 25 to 50 °C were used to screen the optimal pH and temperature, respectively. Absorbance values at a wavelength of 340 nm were recorded using an absorbance microplate reader (Molecular Devices). In these experiments, PBS was used as a control.

Using the same reaction system, the enzyme kinetics were determined under the optimal conditions (pH 7.5, 37 °C), and the experimental data were fitted into the formula through Origin 7.5 [21]:$$V=V_{max}[S]/(K_{m}+[S])\mathrm{or}\frac{1}{V_{0}}=\frac{K_{m}}{V_{max}}$$
where *V_max_* is the maximum velocity, [S] is the substrate concentration, and *K_m_* is the Michaelis-Menten constant.

Similarly, different metal ions (Ca^2+^, K^+^, Na^+^, Fe^3+^ and Mg^2+^) at a different final concentration were added to evaluate the effects on the enzyme activity of EmFBA1.

### 2.8. Data Analysis

Data were presented as mean ± SD (*n* = 3). Origin Pro 7.5 (Northampton, MA, USA) was used to plot, analyze the kinetic data and estimate *K_m_* and *V_max_* using the non-linear fit function.

## 3. Results

### 3.1. Identification of fba Genes in Echinococcus Species

The genome-wide analysis revealed that *E.*
*multilocularis* contained four potential *fba* genes, named as *emfba1**-4*, and there were also four homologues in *E.*
*granulosus* (Figure 1A and Appendix A). Each *fba* had only one copy, and the four *fba* genes in each parasite were distributed on different chromosomes (Figure 1A). Among these *fba* genes, the overall amino acid similarity was 74.64%, whereas each *fba* shared >92% similarity with its corresponding ortholog. The phylogenetic tree showed that these *fba* genes formed four branches, each of which consisted one *fba* of *E.*
*multilocularis* and its corresponding ortholog of *E.*
*granulosus* (Figure 1B). Moreover, all FBA proteins were predicted to contain one domain of FBA I and multiple conserved active site residues (Figure 1C and Appendix A).

### 3.2. Localization of EmFBA1

The purity of EmFBA1 was estimated to be more than 92% (Appendix A). Western blotting results showed that natural EmFBA1 (~40 kDa) could be clearly recognized by the antibodies raised in the study (Figure 2A and Appendix A). As expected, the antibodies also cross-reacted with natural FBA1 from *E*. *granulosus*, *Taenia asiatica*, and *Taenia hydatigena,* suggesting that FBA1 is widely expressed in cestodes. For the localization of EmFBA1, strong immunofluorescent signals were predominantly present along the germinal layer and in the posterior of unevaginated and evaginated protoscoleces (Figure 2B). No signals were observed in the control slides (Figure 2B).

### 3.3. Enzyme Kinetics of EmFBA1

Comparison of EmFBA1 enzyme activity demonstrated that the optimal reaction temperature and pH were 37 °C and 7.5, respectively (Figure 3A,B). Nevertheless, EmFBA1 was still active in a broad range of temperature and pH (Figure 3A,B). The enzyme activity of EmFBA1 was 67.42 U/mg, and it completely converted the substrate within 10 min, while PBS as the control showed no significant difference (Figure 3C). The *K_m_* and *V_max_* of EmFBA1 were 1.75 mM and 0.5 mmol/min, respectively (Figure 3D).

It was further found that Fe^3+^ significantly inhibited the EmFBA1 activity compared with other ions, including Na^+^, Ca^2+^, Mg^2+^ and K^+^ (Figure 3E,F). Moreover, the enzyme activity of EmFBA1 was remarkably abolished with the addition of Fe^3+^ at a final concentration of 0.5 mM (Figure 3F).

## 4. Discussion

Glycolysis is a process of the decomposition of glucose or glycogen into lactic acid and the production of ATP in the presence or absence of oxygen, named as aerobic glycolysis or anaerobic glycolysis. Both *E. multilocularis* and *E. granulosus* can produce energy by glycolysis without a significant difference in the rate of glycogen use under aerobic and anaerobic conditions [22,23]. It has been also shown that glycolysis is critical for the survival of *Trypansoma brucei* during its lifecycle, as it is the only source of ATP. Only 50% glycolysis inhibition is enough to kill *T. brucei*, which makes it a potent target for drugs [24,25]. Therefore, it is plausible to develop the anti-echinococcosis interventions via targeting the glycolysis pathways. FBA, a member of aldolase family, is involved in the glycolysis, gluconeogenesis and Calvin cycle in photosynthesis and widely exists in animals, plants and microorganisms, showing different responses under various stress conditions [26,27,28]. In *S. japonicum*, FBA was potentially involved in the growth and development of sporocysts [13]. In apicomplexan parasites, aldolase acted as a bridge between cell surface adhesion and actin cytoskeletons during invasion [29,30]. In both *E. multilocularis* and *E. granulosus*, all eight FBA shared the same active site residues, which were also reported in the FBA isoenzymes in *Clonorchis sinensis* [31], suggesting that they catalyze the substrates under a similar mechanism. In the phylogenetic tree, each emFBA formed a single branch with its homologue from *E. granulosus*. This may be explained by the fact that a set of emFBA genes might have been formed before speciation of both *Echinococcus* parasites.

It was herein shown that EmFBA1 was widely localized along the germinal layer and in the posterior of the protoscolex. This distribution pattern is consistent with that of FBA1 in *E. granulosus* [12]. As the germinal layer is involved in the growth of cysts and the generation of protoscolex [32], EmFBA1 may play a role in these processes. Moreover, EmFBA1 was only localized in the posterior of both unevaginated and evaginated protoscoleces, suggesting that the posterior is a main venue of anaerobic oxidation of glucose. Previous studies reported that FBA were also localized on the sucker in *Opisthorchis viverrini*, which may provide energy to muscle tissues [33], and on the surface of *Plasmodium* spp., which may be involved in the movement and invasion of parasites [15,34]. Of interest is to investigate the localization of other emFBAs in future studies.

FBA plays a key role in the anaerobic metabolism of glucose. The activity of EmFBA1 was 67.42U/mg, as three times lower as one of *S. japonicum* that has eight different FBAs [18]. This discrepancy may be explained by the fact that the *E. multilocularis* FBA proteins are functionally redundant. As expected, EmFBA1 show the highest enzymatic activity at 37 °C and pH 7.5. In the presence of four ions including Na^+^, Ca^2+^, Mg^2+^ and K^+^, EmFBA1 exhibits no significant alterations in enzymatic activity, suggesting ion independence. It was also shown that EmFBA1 was susceptible to Fe^3+^ ion, which has never been described for other FBAs. This adverse effect of Fe^3+^ is still unclear. Whether other emFBAs have similar traits or not needs to be investigated in future.

## 5. Conclusions

We herein genetically and biochemically characterized FBA1. The results demonstrated that FBA1 was conserved in all cestodes investigated and EmFBA1 showed a relatively lower enzymatic activity. The current study reveals the discrepancy in the biochemical characters of EmFBA1 and is informative for further investigation of its role in the glycolysis in *E. multilocularis*.

## Figures and Tables

**Figure 1 vetsci-09-00004-f001:**
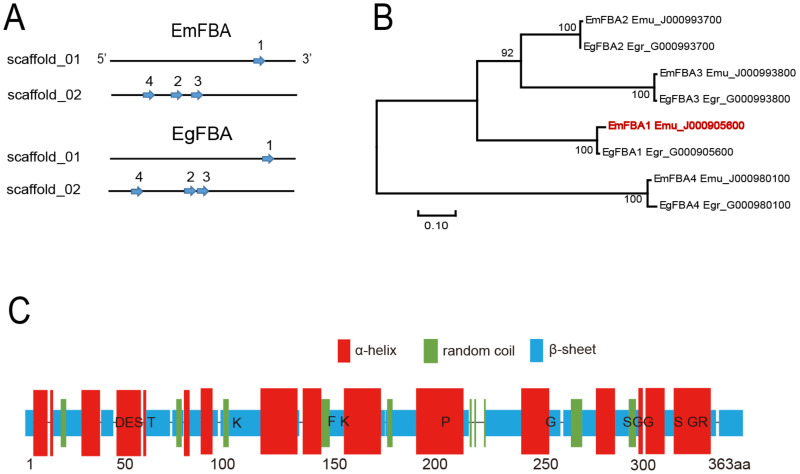
Identification of FBA genes in *Echinococcus* species. (**A**) A set of four different *fba* genes, named *fba1-4*, were distributed on different chromosomes (Scaffold_01/02) of *E. multilocularis* (*EmFBA*) and *E. granulosus* (*EgFBA*). (**B**) A PhyML tree of FBA proteins. The numbers at each node are a bootstrap value greater than 90. (**C**) Predicted secondary structure and conserved active sites of EmFBA1. The predicted conserved active sites were directly indicated. ‘aa’: amino acid.

**Figure 2 vetsci-09-00004-f002:**
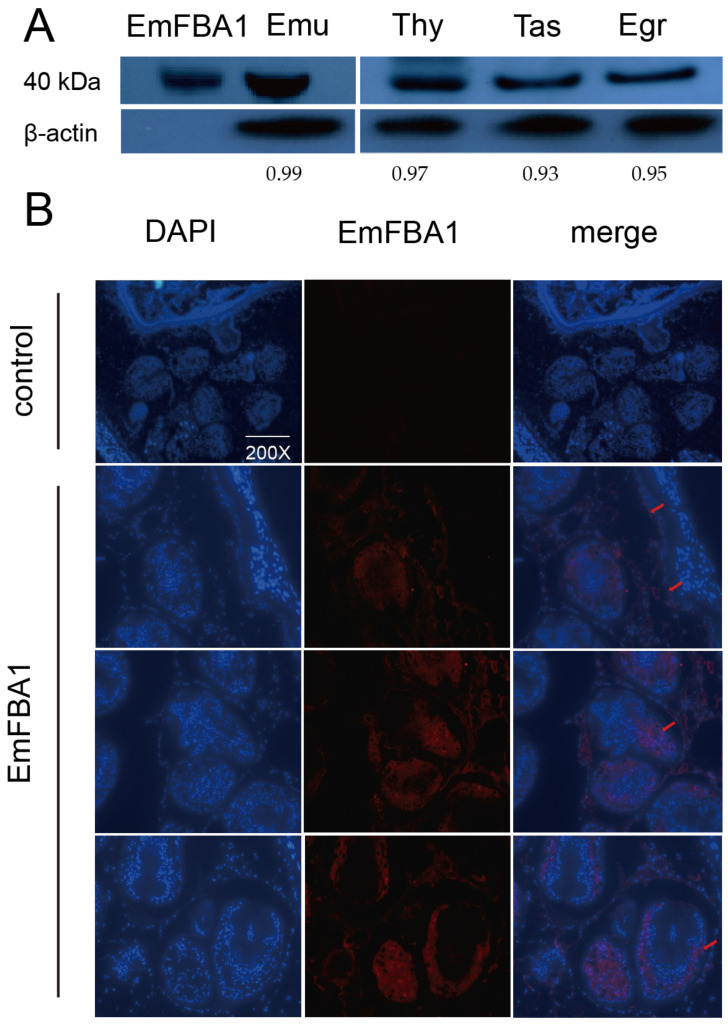
Localization of EmFBA1. (**A**) Western blotting analysis of natural FBA1 in *E. multilocularis* and other tapeworms. Emu, *E. multilocularis*; Egr, *E. granulosus*; Tas, *Taenia asiatica*; Thy, *Taenia hydatigena*. (**B**) Localization of EmFBA1 in *E. multilocularis* cyst. The localization of EmFBA1 was indicated by red arrows. In these experiments, healthy rabbit serum was used as a control.

**Figure 3 vetsci-09-00004-f003:**
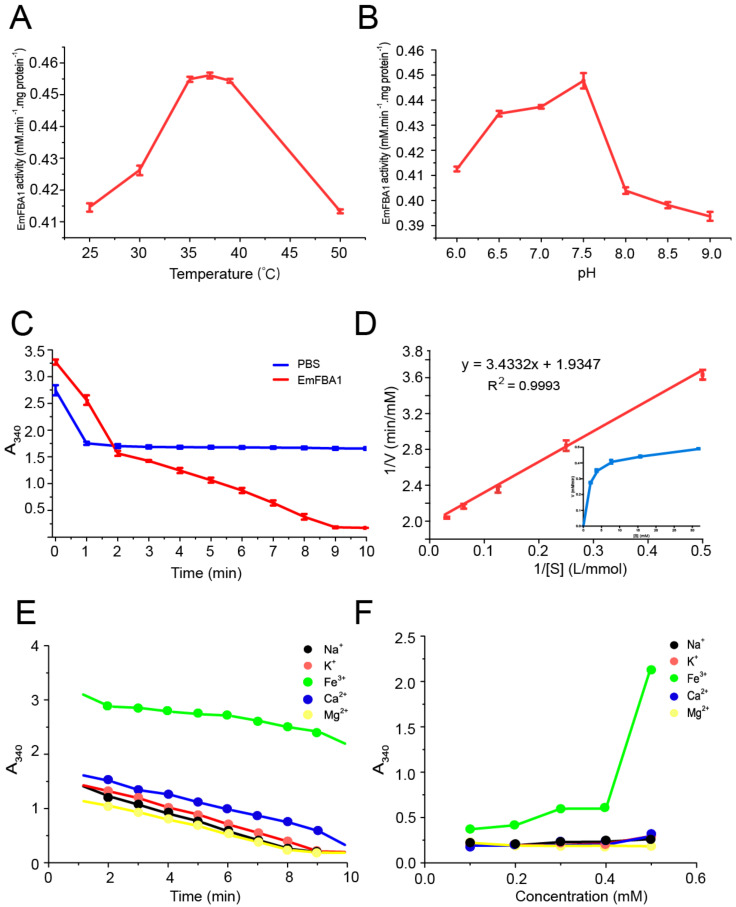
Enzyme kinetics of EmFBA1. (**A**) Effects of temperature on the enzyme activity. At 10 min after reaction, absorbance values at 340 nm were recorded at a different temperature. (**B**) Effects of pH on the enzyme activity. At 10 min after reaction, absorbance values at 340 nm were recorded at a different pH. (**C**) The enzyme activity of EmFBA1. The reaction was conducted in 50 mM Tris-HCl (pH 7.5) at 37 °C for 10 min, and absorbance values at 340 nm were recorded at an interval of 1 min. (**D**) Determination of *K_m_* and *V_max_* of EmFBA1. (**E**) Effects of ions on the enzyme activity of EmFBA1. (**F**) Effects of ions at a different concentration on the enzyme activity of EmFBA1.

## Data Availability

The original contributions presented in the study are included in the article, and further inquiries can be directed to the corresponding authors.

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
