# Peer review of "Characterization of Fructose-1,6-Bisphosphate Aldolase 1 of *Echinococcus multilocularis"

_vetsci, 2021, doi:10.3390/vetsci9010004_

Round 1
Reviewer 1 Report
This research article by Xuedong et al., aimed to conduct genome-wide analysis of the FBA genes in Echinococcus species, and analyzed the secondary structure of E. multilocularis FBA1 (EmFBA1) and its localization by Immunofluorescence assay and determined the enzyme kinetics of EmFBA1.
The aim of this study was an important area and well-designed study. The reviewer has some minor concerns,
- Authors should add loading control beta actin for Western blot.
- Authors should discuss the difference between aerobic glycolysis and anaerobic glycolysis, specify the type of glycolysis in Echinococcus.
- It is not clear, how the enzymatic kinetics translatable, should discuss it clearly.
- Does inhibition of glycolysis can inhibit the growth and multiplication of parasite? or targeting FAB would be clinically important to cure disease? need to discuss in the discussion.
Author Response
Reviewer #1:
This research article by Xuedong et al., aimed to conduct genome-wide analysis of the FBA genes in Echinococcus species, and analyzed the secondary structure of E. multilocularis FBA1 (EmFBA1) and its localization by Immunofluorescence assay and determined the enzyme kinetics of EmFBA1.
The aim of this study was an important area and well-designed study. The reviewer has some minor concerns,
- Authors should add loading control beta actin for Western blot.
Response 1: Thanks for reviewer’s reminder. We have added beta actin as a loading control in Fig. 2A.
- Authors should discuss the difference between aerobic glycolysis and anaerobic glycolysis, specify the type of glycolysis in Echinococcus.
Response 2: Thanks for reviewer’s reminder. The following contents have been added into the Discussion:
‘Glycolysis is a process of the decomposition of glucose or glycogen into lactic acid and the production of ATP in the presence or absence of oxygen, named as aerobic glycolysis or anaerobic glycolysis. Both E. multilocularis and E. granulosus can produce energy by glycolysis without a significant difference in the rate of glycogen use under aerobic and anaerobic conditions [22, 23].’
- It is not clear, how the enzymatic kinetics translatable, should discuss it clearly.
Response 3: Thanks for reviewer’s suggestion. The following contents have been added into Discussion:
‘As expected, EmFBA1 show the highest enzymatic activity at 37℃ and pH 7.5. In the presence of four ions including Na+, Ca2+, Mg2+ and K+, EmFBA1 exhibits no significant al-terations in enzymatic activity, suggesting ion independence. It was also shown that EmFBA1 was susceptible to Fe3+ ion, which has never been described for other FBAs. This adverse effect of Fe3+ is still unclear.’
- Does inhibition of glycolysis can inhibit the growth and multiplication of parasite? or targeting FAB would be clinically important to cure disease? need to discuss in the discussion.
Response 4: Thanks for reviewer’s comments. The following contents have been added into Discussion:
‘It has been also shown that glycolysis is critical for the survival of Trypansoma brucei during its lifecycle, as it is the only source of ATP. Only 50% glycolysis inhibition is enough to kill T. brucei, which makes it a potent target for drugs [24, 25]. Therefore, it is plausible to develop the anti-echinococcosis interventions via targeting the glycolysis pathways.’
Reviewer 2 Report
Authors characterized an important glycolytic enzyme (FBA) in a parasitic cestode, Echinococcus multilocularis. A similar work had been published for a close relative of the parasite, E. granulosus, in 2012. Nevertheless, the present work is of interest for parasitologists, because the paper combines molecular structure studies with enzyme activity studies. It at first shows that Fe3+ ions significantly inhibit FBA activity.
The experiments are carefully done and the manuscript is well written. Nevertheless, some corrections are necessary:
- keywords: replace the semicolon in F1,6B by a comma
- p.2: Schistosoma mansoni is the correct name; immunofluorescence assay in lowercase letter
- p.3: trans-5 cells in lowercase letter
- Results and others: gene names must be written in italics, protein names in normal font
- Fig. 3: I would like to see enzyme activities on the y-axis and not A340 values after 10 min. Then one could for example better see that pH and temperature have optima at 7.5 and 36°C, respectively. Please explain wether SD or SEM of means (how many values?) are shown. I miss any statistical treatment of the data
- References are not written according to MDPI authors' instructions.
Author Response
Reviewer #2: Authors characterized an important glycolytic enzyme (FBA) in a parasitic cestode, Echinococcus multilocularis. A similar work had been published for a close relative of the parasite, E. granulosus, in 2012. Nevertheless, the present work is of interest for parasitologists, because the paper combines molecular structure studies with enzyme activity studies. It at first shows that Fe3+ ions significantly inhibit FBA activity.
The experiments are carefully done and the manuscript is well written. Nevertheless, some corrections are necessary:
- keywords: replace the semicolon in F1,6B by a comma
Response 1: We have revised.
- p.2: Schistosoma mansoni is the correct name; immunofluorescence assay in lowercase letter
Response 2: We have revised.
- p.3: trans-5 cells in lowercase letter
Response 3: We have revised.
- Results and others: gene names must be written in italics, protein names in normal font
Response 4: We have revised.
- Fig. 3: I would like to see enzyme activities on the y-axis and not A340 values after 10 min. Then one could for example better see that pH and temperature have optima at 7.5 and 36°C, respectively. Please explain wether SD or SEM of means (how many values?) are shown. I miss any statistical treatment of the data.
Response 5: Thanks for reviewer’s comments. We have revised the figures 3 A and B and added the subsection of data analysis.
- References are not written according to MDPI authors' instructions.
Response 6: We have revised.

Reviewer 3 Report
Overall happy with the manuscript. A few points to address before acceptance for publication:
- The authors have seemed to miss the latest aldolase paper on Teladorsagia circumcincta. Nematodes are must closer to cestodes so it would be good to compare the properties.
- Please mention the ethics application number.
- 2.5 Methods: Please state the protein expression conditions in detail.
- 2.6. Mention the route of administration and the dose given to rabbits.
- 3.2 whats was the size of aldolase?
Author Response
Reviewer #3: Overall happy with the manuscript. A few points to address before acceptance for publication
- The authors have seemed to miss the latest aldolase paper on Teladorsagia circumcincta. Nematodes are must closer to cestodes so it would be good to compare the properties.
Response 1: Thanks for reviewer’s suggestion. The paper you mentioned (Molecular & Biochemical Parasitology 240 (2020) 111335) is about the malate synthase of Teladorsagia circumcincta. It is an obviously different enzyme and I don’t think it is meaningful for comparison.
- Please mention the ethics application number.
Response 2: We have added.
- 2.5 Methods: Please state the protein expression conditions in detail.
Response 3: Thanks for reviewer’s comments. The following contents have been added into the subsection 2.5 “The positive clone was induced with 1 mM IPTG (BBI life sciences) at a final concentration of 0.5 mmol/L on 20 ℃ for 8 h. The collected bacterial cells were treated by ultrasound, centrifuged at 4 ℃ for 20 min at 10,000×g, and the supernatant were used for purification of EmFBA1.”
- 2.6. Mention the route of administration and the dose given to rabbits.
Response 4: Thanks for reviewer’s comments. The following contents have been added into the subsection 2.6 “First, purified EmFBA1 was mixed with an equal volume of Freund’s complete adjuvant and well emulsified. Then rabbits were subcutaneously injected at multiple locations with emulsified EmFBA1 of 200 μg per rabbit. After 15 days, the second immunization with 200 μg EmFBA1 emulsified with an equal volume of Freund’s incomplete adjuvant was administrated in each rabbit. Finally, the last immunization was performed after 15 days in the same way.”
- 3.2 whats was the size of aldolase?
Response 5: It is about 40 kDa.

This manuscript is a resubmission of an earlier submission. The following is a list of the peer review reports and author responses from that submission.